# Developmental Toxicity Study of DL-4-Hydroxy-4-Phenylhexanamide (DL-HEPB) in Rats

**DOI:** 10.3390/life13081714

**Published:** 2023-08-10

**Authors:** José Melesio Cristóbal-Luna, María Angélica Mojica-Villegas, Sergio Enrique Meza-Toledo, Yuliana García-Martínez, Angélica Pérez-Juárez, Germán Chamorro-Cevallos

**Affiliations:** 1Laboratorio de Toxicología Preclínica, Departamento de Farmacia, Escuela Nacional de Ciencias Biológicas, Instituto Politécnico Nacional, Av. Wilfrido Massieu 399, Col. Nueva Industrial Vallejo, Del. Gustavo A. Madero, Mexico City 07738, Mexico; moviangel13@yahoo.com.mx (M.A.M.-V.); ygarciamart@hotmail.com (Y.G.-M.); 2Laboratorio de Quimioterapia Experimental, Departamento de Bioquímica, Escuela Nacional de Ciencias Biológicas, Instituto Politécnico Nacional, Unidad Profesional Lázaro Cárdenas, Prolongación de Carpio y Plan de Ayala s/n, Col. Santo Tómas, Alcaldía Miguel Hidalgo, Mexico City 11340, Mexico; semeza@hotmail.com; 3Laboratorio de Medicina de Conservación, Escuela Superior de Medicina, Instituto Politécnico Nacional, Plan de San Luis y Díaz Mirón, Col. Casco de Santo Tomás, Del. Miguel Hidalgo, Mexico City 11340, Mexico; apjuarezz@yahoo.com.mx

**Keywords:** DL-4-hydroxy-4-phenylhexanamide, developmental, toxicity, anticonvulsant, rats

## Abstract

Antiepileptic drugs affect embryonic development when administered during pregnancy, generating severe alterations, such as as cleft lip, spina bifida, heart abnormalities, or neuronal alterations. The compound DL-4-hydroxy-4-phenylhexanamide (DL-HEPB), a phenyl alcohol amide structurally different from known anticonvulsants, has shown good anticonvulsant effects in previous studies. However, its effects on intrauterine development are unknown. So, the purpose of this study was to determine the potential of DL-HEPB to produce alterations in conceptus. Pregnant Wistar rats were orally exposed to 0, 50, 100, and 200 mg/kg of DL-HEPB during organogenesis, and their food consumption and weight gain were measured. On gestation day 21, pregnant females were euthanized to analyze the fetuses for external, visceral, and skeletal malformations. A significant decrease in food consumption and body weight was observed in mothers, without any other manifestation of toxicity. In fetuses, no external malformations, visceral, or skeletal abnormalities, were observed under the dose of 100 mg/kg, while the dose of 200 mg/kg caused malformations in low frequency in brain and kidneys. In view of the results obtained, DL-HEPB could be a good starting point for the design of new highly effective anticonvulsant agents, with much lower developmental toxicity than that shown by commercial anticonvulsants.

## 1. Introduction

Epilepsy was considered to be “a disorder of the brain characterized by an enduring predisposition to generate epileptic seizures and by the neurobiological, cognitive, psychological, and social consequences of this condition” [1] affecting 50 million persons worldwide [2]. This disorder represents one of the most common chronic diseases, causing high rates of mortality, comorbidity, and disability in underdeveloped countries [3], the risk of early death in people with epilepsy being three times higher than in the general population [4].

Three generations of anticonvulsant drugs (AED) are known, which have been classified depending on their time of appearance on the market: the first generation, 1857 to 1958, included molecules derived from barbiturate (primidone, phenytoin, ethosuximide and trimethadione), phenobarbital, and potassium bromide. The second-generation, launched between 1960 and 1975, differed chemically from the barbiturates and included molecules such as carbamazepine, valproate, and benzodiazepines. Finally, the third-generation of AED designed to selectively target a mechanism thought to be critical for the occurrence of epileptic seizures was launched in the 1980s, and includes sulfonamides, thiadiazoles, semi- and thiosemicarbazones, pyrrolidine-2,5 diones, imidazoles, benzothiazoles, and amino-acid derivatives [5,6]. These drugs possess some pharmacokinetic advantages, such as good bioavailability, plasma–protein binding, and fewer drug–drug interactions than other anticonvulsants [7]. Several of these current AED have been associated with severe side effects, such as hepatotoxicity produced by valproic acid, phenytoin and felbamate and developmental toxicity [8]. Today, the most used AED to control seizures are sodium valproate, carbamazepine, clonazepam, phenytoin, levetiracetam, and topiramate [9]. Many of these molecules share their action through Ca^2+^, Na^+^ channels, synaptic vesicle protein SV2A, or GABAergic transmission [10]; its exposure during intrauterine development increases the risk of major congenital malformations two- to three-fold, and the adverse cognitive outcomes in the offspring. The specific malformations that these drugs induce are very wide, so the most common harmful effects on embryofetal development are congenital heart disease, cleft lip/palate, urogenital defects, neural tube defects, intrauterine growth restriction, impaired cognitive development, and adverse behavioral effects [11,12,13]. In this regard, valproate has been the drug with the highest risk of developing alterations in embryofetal development, followed by lamotrigine, levetiracetam, and oxcarbazepine alterations, which limit their clinical use [14].

It has been calculated that approximately one half of patients with epilepsy are women; therefore, the treatment can often coincide with pregnancy [13]. Thus, many embryos and fetuses may be exposed to anticonvulsant drugs [15,16]. Several scientific works demonstrated that AED therapy, rather than maternal disease or convulsions, is the cause of some of these disorders [17], although it is probable that different factors are likely to be contributors, such as genetically determined susceptibility [18]. Status epilepticus is a clinical problem in medicine. Nearly 30% of epileptic patients exhibit resistance to the treatment. Phenyl alcohol amides are the only anticonvulsant drugs known that protect against the gamma-aminobutyric acid withdrawal syndrome (GWS), a model of epilepsy which is resistant to clinical antiepileptic drugs such as phenytoin, barbiturates, ethosuccimid, valproic acid, carbamazepine, diazepam (the drug choice in cases of status epilepticus), and even pentobarbital at anesthetic doses [19]. Therefore, it is necessary to intensify our efforts to develop chemical structures other than the conventional AED, to generate new molecules with greater efficiency and less toxicity that help prevent possible effects on development and maintain the health of the pregnant mother [20].

The compound DL-4-hydroxy-4-phenylhexanamide (DL-HEPB) is a molecule that in previous studies has protected cats and rats against epileptogenesis induced by hippocampal kindling, and mice against seizures induced by maximal electroshock, pentylenetetrazol, bicuculline, 4-aminopyridine, and thiosemicarbazide [21,22,23]. Despite being a molecule with notable anticonvulsant effects, it does not have preclinical toxicology studies that allow it to advance to clinical trials. However, DL-HEPB is positioned as a good alternative in the development of anticonvulsant molecules because it has a chemical structure that is very different from other anticonvulsants on the market (Figure 1) [24]. Currently, promising results have been found for this molecule and some of its derivatives [25,26].

The structural similarities between DL-HEPB and certain GABAB receptor ligands suggested that this compound could antagonize GABA actions at the GABAB receptors. However, electrophysiological, and neurochemical evidence is still missing (it is only a theory that we must test). So, its profile propose that it could be useful against mild types of epilepsy of the absence type [27].

Considering the wide anticonvulsant effect of DL-HEPB and the scarcity of available information on its teratogenicity, regardless of studies already conducted on mice [28], this research was carried out to evaluate its potential to affect fetal development throughout pregnancy in rats.

## 2. Materials and Methods

### 2.1. Animals and Housing Conditions

In total, 10 sexually mature male and 90 female Wistar rats aged 10 weeks were obtained from the breeding colony of the Centro de Investigaciones y de Estudios Avanzados del Instituto Politécnico Nacional (CINVESTAV), Mexico City. The rats were housed in polycarbonate cages with pinewood shavings as bedding in an air-conditioned room under controlled conditions (22–23 °C, 55–60% relative humidity, and artificial illumination between 8:00 and 20:00 h). The animals had access to Purine Rodent Chow and tap water ad libitum and were acclimatized for 2 weeks prior to the initiation of the experiment.

Approval for the study was obtained from the Comité de Bioética de la Escuela Nacional de Ciencias Biológicas under protocol number 09-CEI-002-20190327 and according to the Mexican Standard (NOM ZOO-062-00-1999) concerning technical statements for production, care, and use of experimental laboratory animals.

### 2.2. Oral Acute Toxicity

An acute oral toxicity study was performed during 2 weeks in 10 female Wistar rats to calculate the LD_50_ of DL-HEPB by the “up-and-down procedure” recommended by the Organization for Economic Co-operation and Development [29], in Test No. 425: Acute Oral Toxicity: up-and-down procedure. This is in order to incorporate the “3R” principles of animal research to evaluate the toxicity, since it uses a smaller number of experimental animals, and provides accurate results, and is significantly simpler than traditional methods such as those of Karber, Lorke, Miller, and Tainer [30,31,32].

### 2.3. Teratogenic Study

For our study, groups of three female rats and one sexually matured male rat were caged together overnight; on the following morning, copulation was confirmed by vaginal smears, and the day of spermatozoa detection or vaginal plug was considered gestational day 0 of pregnancy (GD0). The pregnant females were separated, housed into individual cages, and randomly distributed into the four experimental groups who integrated our experiment (*n* = 20). DL-HEPB was suspended in the vehicle (water containing 1% of polysorbate 80 Sigma-Aldrich) and was administered daily by gastric intubation at dosage levels of 0 (control group (vehicle)), 50, 100, and 200 mg/kg at a constant volume of 10 mL/kg. Dosing suspensions for treatments were prepared each day and the rats were treated with the vehicle or with DL-HEPB on GD6-16. The doses used were selected from the result obtained in our acute toxicity study: the dose of 200 mg/kg was selected because it is approximately one tenth of the LD_50_ value obtained in our study (1886.4 mg/kg), while the doses of 100 and 50 mg/kg correspond to half and a quarter of this dose, respectively. In addition, previous studies that evaluated the anticonvulsant effect of DL-HEPP, an analogue of DL-HEPB, where good anticonvulsant effects were found at the 50 mg/kg dose [26] and anticonvulsant effects like those of valproate at a dose of 100 mg/kg [21]. For its part, the dose of 200 mg/kg was selected as twice the highest dose tested, with the intention of ensuring the observation of the possible toxic effects on development that DL-HEPB could generate. During gestation, the animals were monitored for food consumption and weight gain, abnormalities of condition or behavior, and mortality and morbidity on GD0, 6, 16, and 20. The quantification of food intake was performed manually: a known amount of food was placed in each of the cages in which the pregnant females were housed individually. After 24 h, rats were removed from their cages and the amount of food remaining was recorded (that of the feeder plus crumbs that fell on plastic sheets placed under the feeder or on the bottom of the cage). Intake was calculated as the weight of food (g) provided less that recovered [33].

On GD20, the pregnant females were euthanized by CO_2_ asphyxiation and cervical dislocation. Gravid uterus weights were measured and further examined for the number of corpora lutea, implantations, and resorption sites. The number of live fetuses per litter, dead young, fetal weights, placental weights, crown–rump females and males, female and male anogenital distance, sex ratio, and external abnormalities of fetuses were registered.

Selected organs (liver, lung, brain, heart, and kidneys) were observed macroscopically in the dams without finding alterations in any of them. Prior to sacrifice by hypothermia in cold physiological solution, two thirds of fetuses in each litter were fixed in Bouin’ solution for the study of visceral malformations using the Wilson and Warkany (1964) [34] free-hand slicing technique, while the remaining one third of fetuses was investigated to reveal skeletal and cartilage abnormalities after being skinned, eviscerated, fixed in 95% ethanol, cleared with potassium hydroxide, and tinged with alizarin red-S and alcian blue according to the Peters double-staining method [35]. The skeletons were scrutinized under a stereomicroscope to assess the degree of ossification, the presence/absence of bones, bone deformation, fused bone, degree of ossification, and the presence of additional elements such as rudimentary ribs [36].

### 2.4. Statistical Analysis

Statistical analyses were performed employing the litter as the experimental unit [37]. All data were analyzed using the Shapiro–Wilk test to determine if they followed a normal distribution. Next data variables such as food consumption, maternal body weight, reproductive performances, and fetal developmental parameters were assessed by analysis of variance (ANOVA) and post hoc Student–Newman–Keuls. Live and dead fetuses, resorption and sex ratios, number of litters with live fetuses or resorptions, fetal skeletal variations, and fetuses with external, visceral, and skeletal findings were compared utilizing Chi square and Fisher exact tests. Statistical significance was set at *p* < 0.05.

## 3. Results

### 3.1. Acute Oral Toxicity

The result of acute oral toxicity evaluation revealed an LD_50_ of 1886.4 mg/kg, which indicates a moderate toxicity for DL-HEPB (category 4), according to the classification criteria for acute oral toxicity or acute toxicity estimate (ATE) (300 < LD_50_ < 2000 mg/kg) of the Globally Harmonized System of Classification and Labelling of Chemicals (GSH) [38]. It should be noted that before their death, administered animals showed agitation, diarrhea, and abdomen contractions.

### 3.2. Teratogenic Study

In the pregnant rats of our developmental toxicity study, there was no DL-HEPB treatment-related mortality. However, at dose of 200 mg/kg there was a significant decrease (*p* < 0.05) in food consumption during the treatment period (GD6-GD15) and during the post-treatment period (GD16-GD20), compared to the control group, with values of 22.5 ± 3.6 and 29.2 ± 4.2 g, respectively. A lower maternal weight was also found to be significantly different (*p* < 0.05) during these same periods with values of 31.2 ± 11.4 and 46.9 ± 15.3 g in relation to the same group (Table 1). Upon euthanasia, the treatment with DL-HEPB, even at the dose of 200 mg/kg, did not affect the macroscopic aspect of the lung, brain, and heart of pregnant females. However, the liver exhibited a dark red color, and in the kidneys, there were some dark spots.

As depicted in Table 2, except for litters with resorptions and resorptions per litter at a dose of 200 mg/kg of DL-HEPB, none of the other parameters demonstrated a significant difference (*p* < 0.05) with the control group. Consequently, the increased number of resorptions found in uterus showed that DL-HEPB produced embryotoxicity.

Table 3 presents the developmental parameters of viable fetuses. No effects of DL-HEPB on fetal and placental weights were observed. No statistically significant differences were found, either (*p* < 0.05), in crown–rump length or in anogenital distance between fetuses of the same sex and different treatments. There were no significant changes in the sex ratio male/female found in the litters.

Table 4 presents the results of the external and visceral examinations of the fetuses. Some malformations were found in the negative control and in the three DL-HEPB-treated groups. None of these malformations, except for hydrocephalus and hydronephrosis, appear to be due to the treatment. It should be noted that they occurred only in the highest dose with low frequency. Other fetal alterations, such as the small hematomas that were observed in any dosage group, were unremarkable.

The results of the skeletal analysis of the fetuses are presented in Table 5. The variations found by the administration of the different doses of DL-HEPB and by the administration of the vehicle to the control group were found in few fetuses. The main variations were observed in the skull, vertebrae, sternebrae, tail, and ribs. In all cases, these variations were comparable across all dosage groups with DL-HEPB and the control group.

## 4. Discussion

The objective of this work was to evaluate the effect of DL-HEPB on the embryonic development of Wistar rats, to propose a molecule with good anticonvulsant effects and minimal toxic effects for intrauterine development, which serves as a basis for developing and studying new safe antiepileptic molecules.

Orally administered DL-HEPB did not produce mortality in females, although it produced some signs of toxicity such as decreased food consumption and weight loss during organogenesis (GD6-GD15) and the post-treatment period (GD16-GD20) at a dose of 200 mg/kg during these same periods. The decrease in food consumption and weight gain is probably due to DL-HEPB activity at the level of the ventromedial hypothalamus, which serves as a satiety center: hunger, blood glucose levels, and certain levels of hormones in the body are controlled by it [39]. In addition, dosages up to the highest (200 mg/kg), did not affect any of the organs in the external analysis of fetuses or any clinical signs of toxicity in the females, results consistent with those found in studies previously carried out on mice [28]. This has not been the case with other anticonvulsant agents that, regardless of their good pharmacological activity, have given rise to toxic effects in the liver and kidney of pregnant rats [40] as well as malformations in embryos of rats and chicks [41,42]. In this scenario, the only aspect that we can highlight is the dark coloration observed in the liver of pregnant females. Although this change in liver coloration was partially observed in all groups, including the control rats, it was slightly more evident in females that received high doses of DL-HEPB. Therefore, we do not consider that it is the result of a hepatotoxic effect such as that induced by anticonvulsant agents such as valproate, in which, in addition to the dark color of liver, the injury is characterized by marked steatosis and inflammation, increased oxidative stress, serum transaminases and alkaline phosphatase, among others [43]. In this study, we did not perform any of these enzymatic determinations during the macroscopic evaluation of the livers of females treated with DL-HEPB; however we did not observe signs of steatosis or inflammation that referred to some type of liver damage such as those mentioned. In contrast, it is reported that liver damage generally occurs due to deficiencies in animal nutrition that disrupt normal metabolism [44]. Therefore, we consider that the change in color in the livers treated with 200 mg/kg of DL-HEPB is best explained through changes in metabolism promoted by reduced feed intake due to an action in the arcuate hypothalamus that led to a state of malnutrition [45], rather than through liver damage such as that caused by classical anticonvulsant agents. Something similar could occur with the external alterations observed in the kidneys: nutritional deficiencies associated with reduced food intake have been shown to induce kidney damage in rats characterized by proteinuria, increases in serum creatinine and urea, and changes in the external appearance of the kidneys [46]. Although the observed organ changes were slight, they seem to be associated with nutritional problems rather than direct toxic effects of DL-DEPB.

DL-HEPB was administered up to a 200 mg/kg daily oral dose. During the course of GD6-GD15 period, this did not significantly alter the percentage of post-implantation and other indicators such as the weight of the gravid uterus, which are part of the observations made in the females during sacrifice. However, it did cause an increase in litters with a significant number of resorptions, and therefore in resorptions per litter (*p* < 0.05), a lower number of live fetuses, and associated placentae. These results possibly produced a decrease, although not significant, in the gravid uterine weight, with the two high doses. The majority of the external, visceral, and minor skeletal variations were no more frequent in the treated group than in the control group, with the exception of fetuses presenting hydrocephalus and hydronephrosis, which differed significantly (*p* < 0.05) from the control group with the highest dose of 200 mg/kg. This suggests that in the teratogenic evaluation, DL-HEPB is safe up to the dose of 100 mg/kg and that the alterations observed in the study at lower doses were incidental and devoid of toxicological significance, as compared with other anticonvulsant drugs that induce side effects, principally malformations of the neural tube in animals [47]. The low frequency of alterations found in all the groups could be considered variations and not directly related to the treatments. Hydrocephalus is defined as an increase in the volume of cerebrospinal fluid raised. For several years it has been known that in animals such as Wistar rats, this condition may be due not only to teratogens but also to factors commonly arising from inflammatory processes, expanding lesions (hereditary tumors), and congenital malformations [48]. Although on average, a incidence rate of 0.3% in hydrocephalus is considered normal, in recent years it has been shown that Wistar rats have a certain susceptibility to present ventricular-cerebral and vascular dilatations in different magnitudes with a relatively high incidence (43.2%). Neuroimaging techniques have shown that under normal conditions, ventricular dilatation in this particular strain can occur up to 19.1% [49]. This does not mean than about forty percent of the offspring of Wistar rats present severe spontaneous hydrocephalus, but rather that through neuroimaging it has been possible to analyze in greater detail the development of some brain structures under normal conditions, reaching the conclusion that there may be different degrees of said alteration with a considerable prevalence. Is important to note that this condition has been closely related to a transmembrane protein 67 (TMEM67) mutation, thanks to which a hydrocephalus model has been implemented in Wistar rats [50]. In the same sense, although the developmental defects of the kidney are uncommon in rats, these are not null. Conditions such as cortical, medullary, or papillary cysts, and hydronephrosis are the most frequently observed [51]. Hydronephrosis is a condition characterized by the alteration of micro (atrophy of the renal glomeruli, tubules, and pelvic urothelium) and macro (loss of renal medullary tissue, and dilation of the renal pelvis) structures. Furthermore, it has been estimated that the spontaneous incidence rate of hydronephrosis in most strains of rats (including Wistar) is <2–6% [52]. Under these considerations, the incidence of hydrocephalus and hydronephrosis found in our study could seem normal. However, since these occurred in a very low proportion in the control group, we assume that there is a dose-dependent mechanism that is significantly increasing the frequency of these alterations at doses ≥ 200 mg/kg of DL-HEPB. Hence, it is necessary to carry out more toxicity studies to truly ensure that DL-HEPB does not show dose-associated alterations, if it acts at the level of the TMEM67 protein, or if these alterations are related to a nutritional deficiency driven by the decrease in food consumption, etc. For the aforementioned, the significance of this study lies in the need to develop new anticonvulsant molecules with less toxic effects for embryo-fetal development. DL-HEPB is an excellent candidate that could be outlined to advance in preclinical tests, since in addition to having shown excellent anticonvulsant effects and minimal toxic effects for intrauterine development, it presents a novel structure that opens the door to the development of research that promotes the investigation of other structural analogues. However, there are limitations in the present study that should be investigated in other future works to better understand the toxic effects of DL-HEPB: investigate the receptors on which it exerts its anticonvulsant effect, determine the brain areas involved in its antiepileptic effects and decrease in food intake, evaluate if it has any effect on the behavior or memory of animals exposed in subacute studies, and carry out studies in other species such as fish or flies, to guarantee its safety and facilitate its extrapolation to humans.

## 5. Conclusions

Based on the results of this study, it is concluded that DL-HEPB administered at up to 100 mg/kg/day to pregnant Wistar rats during organogenesis did not induce any evidence of developmental toxicity. However, at 200 mg/kg DL-HEPB could be considered teratogenic.

The absence of developmental effects attributable to DL-HEPB below the dose of 100 mg/kg appears to be consistent with observations for other selected hydroxyamides in other reproductive studies [53]. Additional studies must be conducted to verify the safety of DL-HEPB, which, based also on the pharmacological studies carried out so far, appears to be a promising molecule.

## Figures and Tables

**Figure 1 life-13-01714-f001:**
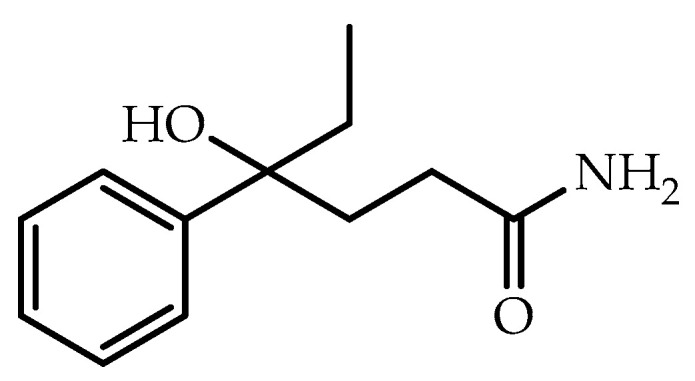
Structure of DL-4-hydroxy-4-ethyl-4-phenylbutiramide (DL-HEPB).

**Table 1 life-13-01714-t001:** Mean of food consumption and maternal body weight in rats administered with DL-HEPB from GD6 to GD15.

Days	Doses of DL-HEPB (mg/kg/day)			
	0 (Control)	50	100	200
Maternal mean food consumption (g) on GD:				
0–5	21.2 ± 2.3	21.6 ± 3.2	20.8 ± 2.9	20.4 ± 3.0
6–15	26.3 ± 2.7	26.5 ± 3.6	25.3 ± 3.4	22.5 ± 3.6 *
16–20	34.1 ± 3.8	30.9 ± 3.9	31.0 ± 3.5	29.2 ± 4.2 *
Maternal weight gain (g) on GD:				
0–5	15.5 ± 5.2	17.8 ± 3.3	18.2 ± 6.1	16.7 ± 5.5
6–15	36.1 ± 8.6	39.4 ± 9.9	34.0 ± 10	31.2 ± 11.4 *
16–20	50.8 ± 11.8	50.0 ± 12.2	48.1 ± 13	46.9 ± 15.3 *

The food consumption resulting from 20 pregnant rats per group. Values are given as the mean ± SEM. DL-4-hydroxy-4-phenylhexanamide, DL-HEPB; GD, Gestational day. * Significant difference (*p* < 0.05) vs. control group.

**Table 2 life-13-01714-t002:** Reproductive performances in rats administered DL-HEPB from GD6 to GD15.

Parameters	Doses of DL-HEPB (mg/kg/day)			
	Control	50	100	200
No. of mated females	20	20	20	20
No. pregnant females	19	18	20	17
Fertility index (%)	95	90	100	85
Pregnancy index (%)	90	90	100	85
Corpora lutea	14.8 ± 1.8	14.2 ± 2.0	13.6 ± 1.9	13.1 ± 2.1
Implantations	13.3 ± 1.6	12.5 ± 2.0	11.9 ± 1.2	11.7 ±1.8
Pre-implantation loss (%)	10.1 ± 12.1	11.0 ± 10.9	12.2 ±14.5	10.3 ±11.6
Post-implantation loss (%)	9.14 ± 14.5	8.8 ± 15.81	8.4 ± 19.6	7.5 ± 16.5
Females with live fetuses	18	18	20	17
No. of live fetuses	217	205	218	184
Live fetuses per litter	12.1 ± 3.2	11.4 ± 3.9	10.9 ± 4.1	10.8 ± 3.5
Dead fetuses	0	0	1	3
Litters with resorptions	5	7	8	14 *
Resorptions per litter	1.5 ± 1.8	1.0 ±1.9	2.6 ±1.2	11.7 ± 4.2 *
Gravid uterine weight (g)	56.6 ± 19.5	55.5 ± 16.8	47.4 ±19.6	45.9 ±18.8

The fetuses resulting from 20 pregnant rats per group were examined. Data are expressed as the mean ± SEM (*n* = 20). DL-4-hydroxy-4-phenylhexanamide, DL-HEPB. Values are given as the mean ± SEM. * Significant difference (*p* < 0.05) vs. control group. Pregnancy index = (No. of females with live fetuses/No. of mated females) (100). Fertility index = (No. of pregnant females/No. of mated females) (100). Post-implantation loss = (No. of implantations − No. of live fetuses/No. of implantations) × (100).

**Table 3 life-13-01714-t003:** Fetal developmental parameters in fetuses of rats administered with DL-HEPB from GD6 to GD15.

Parameters	Doses of DL-HEPB (mg/kg/day)			
	Control	50	100	200
Total examined fetuses	217	205	218	184
Fetal weight (g)	3.16 ± 0.33	3.24 ± 0.34	3.19 ± 0.28	2.95 ± 0.31
Placental weight (g)	0.51 ± 0.12	0.49 ± 0.10	0.48 ± 0.11	0.45 ± 0.10
Females crown–rump length (mm)	3.25 ± 2.5	3.15 ± 2.4	3.20 ± 2.6	3.02 ± 2.6
Females anogenital distance (mm)	2.2 ± 0.3	2.3 ± 0.6	2.2 ± 0.5	2.4 ± 0.6
Males crown–rump (mm)	32.3 ± 2.2	33.1 ±1.8	33.5 ±2.1	30.2 ± 2.2
Males anogenital distance (mm)	3.5 ± 0.5	3.6 ±0.5	3.7 ± 0.6	3.4 ± 0.4
Sex ratio male/female fetuses	0.81	1.18	1.06	1.21

Evaluations of the fetuses from 20 pregnant rats per group. Values are given as the mean ± SEM.

**Table 4 life-13-01714-t004:** External and visceral examinations in fetuses of rats administered with DL-HEPB from GD6 to GD15.

Parameters	Doses of DL-HEPB (mg/kg/day)			
	Control	50	100	200
Total examined fetuses	145	137	146	123
External:				
Gastroschisis	1 (1)	0	0	2
Cephalocele	0	0	1 (1)	0
Anophthalmos	0	1 (1)	0	1 (1)
Microphthalmos	0	1 (1)	0	2 (2)
Cleft lip	0	0	1 (1)	2 (2)
Cleft palate	0	0	0	1 (1)
Hemorrhagic spots	7 (4)	5 (4)	10 (7)	9 (6)
Visceral:				
Hydrocephalus	1 (1)	0	3 (3)	6 (4) *
Ureters dilated	1 (1)	1 (1)	0	2 (1)
Hydronephrosis	1 (1)	1 (1)	3	7 (5) *
Renal pelvis dilated	0	1 (1)	0	1 (1)
Renal ectopic	0	0	1 (1)	2 (2)
Testis displaced	0	1 (1)	0	2 (1)

Evaluations of the fetuses from 20 pregnant rats per group. Values are given as frequencies. Values in parentheses represent the litters (mothers) in which these variations were observed. * Significant difference (*p* < 0.05) vs. control group.

**Table 5 life-13-01714-t005:** Skeletal variations in the fetuses of rats administered DL-HEPB during GD6-GD15.

	Doses of DL-HEPB (mg/kg/day)			
	Control	50	100	200
Total examined fetuses	72	68	72	61
Ossification delays				
Parietal	4 (3)	6 (4)	5 (6)	7 (6)
Inter-parietal	7 (5)	7 (5)	6 (3)	8 (6)
Supraoccipital	1 (1)	3 (2)	2 (1)	4 (3)
Hyoid bones	2 (2)	1 (1)	2 (2)	1 (1)
Cervical vertebrae	3 (2)	5 (3)	7 (3)	7 (5)
Vertebrae, bipartite ossification center	5 (3)	6 (4)	7 (5)	5 (4)
Sternebrae, poor ossification	2 (1)	4 (3)	3 (2)	1 (1)
Tail poor ossifications	10 (4)	6 (2)	11 (7)	13 (8)
Ribs variations				
Short	2 (1)	1 (1)	3 (3)	4 (4)
Wavy	1 (1)	0	1 (1)	2 (1)
Fused	2 (2)	2 (2)	1 (1)	3 (3)
Supernumerary	4 (3)	5 (4)	6 (4)	4 (2)
Absent	0	0	0	1 (1)

Evaluations of the fetuses from 20 pregnant rats per group. Values are given as frequencies. Values in parentheses represent the litters (mothers) in which these variations were observed.

## Data Availability

The data presented in this study are available on request from the corresponding author.

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
