# Peer review of "Developmental Toxicity Study of DL-4-Hydroxy-4-Phenylhexanamide (DL-HEPB) in Rats"

_life, 2023, doi:10.3390/life13081714_

Round 1

Reviewer 1 Report

Manuscript LIFE-2482861

Cristóbal-Luna et al. present a study on the characterization of toxicity of DL-4-hydroxy-4-phenylhexanamide (DL-HEPB), an antiepileptic drug, in pregnant rats. The authors described the treatment with DL-HEPB (at 50, 100, and 200 mg/kg). The results showed a significant decrease in food consumption and body weight in mothers without any other manifestation of toxicity. In fetuses, no external malformations, visceral or skeletal abnormalities were observed until 100 mg/kg. Further, the authors suggested that DL-HEPB (until the dose of 100 mg/kg) would be a good starting point for designing a new highly effective anticonvulsant agent. The manuscript is well written, the methodology is precise and detailed, and the discussion is based on current literature considering the study findings. However, some points need to be clarified before this manuscript is accepted for publication.

Specific comments

1. In line 45, the authors describe three generations of anticonvulsant drugs but do not mention which drugs they are. In the following sentence, they describe the latest generation... This sentence is "lost" and could be rephrased to be more specific; they are classified depending on their time of appearance on the market. Is this enough to describe anticonvulsant drugs?

2. In the Introduction section, there is a description of DL-HEPB treatment for epilepsy, lines 66-74. Are there descriptions of side effects? Is this compost already in a clinic? It is unclear whether this anticonvulsant agent is already in clinical use.

3. There is a description in lines 78-81 of the mechanism of action of DL-HEPB and then in lines 222-224. However, a more detailed description of the mechanisms of action of this compound would be essential to enrich the discussion. Notably, there is a citation of its action via GABAB receptors and that it acts on the ventromedial hypothalamus. What are the actions of the ventromedial hypothalamus? Inhibitory action? Direct? Via which mechanism? The Introduction and Discussion could be improved considering the neuropharmacological aspects of these two points.

4. In the methodology description, lines 109-137 describe the use of the compound to assess the teratogenic effect. However, it does not describe or cite the rationale for the doses used. Have the chosen doses already been used in epilepsy studies? Do they present satisfactory results for the control of epileptic seizures? It is necessary to inform the reference for the chosen doses.

5. In the statistical analysis (lines 139-146), was a test to assess the normality of the data performed so that parametric or non-parametric analysis could be performed? This information needs to be described.

Minor points

On the abstract (line 31), there is an inconsistency "the dose of 400 mg/kg" Shouldn't it be "the dose of 200 mg/kg"?

Author Response

Comment 1: In line 45, the authors describe three generations of anticonvulsant drugs but do not mention which drugs they are. In the following sentence, they describe the latest generation... This sentence is "lost" and could be rephrased to be more specific; they are classified depending on their time of appearance on the market. Is this enough to describe anticonvulsant drugs?

Response 1: Thank you for your observation. We agree with your comment as it enriches the content of the manuscript. The information has been added in the corresponding section (lines 46-52).

Comment 2: In the Introduction section, there is a description of DL-HEPB treatment for epilepsy, lines 66-74. Are there descriptions of side effects? Is this compost already in a clinic? It is unclear whether this anticonvulsant agent is already in clinical use.

Response 2: Thanks for your comment. The description of the antiepileptic effects of DL-HEPB that you mention refers to the studies that have been carried out to evaluate the protective effect of this molecule in cat, rat, and mouse animal models, in which seizures were induced by different models of epilepsy. Therefore, HEPB is not available on the market nor does it have clinical studies. This document is the first investigation in preclinical phases of the toxicity of DL-HEPB. Hence its importance. In the antiepileptic studies the administration of DL-HEPB has been unique and has not shown side effects. Therefore, the only side effects of this molecule are those described in the present document after repeated administrations. We have made some changes to clarify that DL-HEPB is not in research or clinical use (lines 88-90).

Comment 3: There is a description in lines 78-81 of the mechanism of action of DL-HEPB and then in lines 222-224. However, a more detailed description of the mechanisms of action of this compound would be essential to enrich the discussion. Notably, there is a citation of its action via GABAB receptors and that it acts on the ventromedial hypothalamus. What are the actions of the ventromedial hypothalamus? Inhibitory action? Direct? Via which mechanism? The Introduction and Discussion could be improved considering the neuropharmacological aspects of these two points.

Response 3: Thanks for your comment. Your question is very interesting, but unfortunately it does not exist an established mechanism to explain the antiepileptic effects of DL-HEPB. In order to elucidate its neuropharmacological mechanism, it is necessary to carry out several studies that suggest us determine the receptors or brain structures in which it is acting. As you rightly point out, in lines 97-101 we suggest that DL-HEPB could antagonize GABA actions at the GABAB receptors, due to the structural similarities between DL-HEPB and certain GABAB receptor ligands. However, electrophysiological and neurochemical evidences are still missing (it is only a theory that we must test). The same is true in the discussion section on lines 256-259; due to the observed weight loss and our theory of DL-HEPB structure, we think that it could act on this regulatory center of satiety since hunger, blood glucose levels, and certain levels of hormones in the body are controlled for the hypothalamus. So, if we do not have solid elements that allow us to propose a mechanism of action, much less we have elements to try explaining what are the actions of the ventromedial hypothalamus that are affected or inhibited direct or indirect way...? It would be irresponsible of us to propose something like that. However, a point was made in the corresponding section (lines 258-259).

Comment 4: In the methodology description, lines 109-137 describe the use of the compound to assess the teratogenic effect. However, it does not describe or cite the rationale for the doses used. Have the chosen doses already been used in epilepsy studies? Do they present satisfactory results for the control of epileptic seizures? It is necessary to inform the reference for the chosen doses.

Response 4: Thank you for your insightful observations. After analyzing your recommendation, we agree that it is important to include the data you mention. The doses used were selected from the result obtained in our acute toxicity study: the dose of 200 mg/kg was selected because it is approximately one-tenth of the LD50 value obtained in our study (1,886.4 mg/kg), while the doses of 100 and 50 mg/kg correspond to half and a quarter of this dose, respectively. In addition, previous studies that evaluated the anti-convulsant effect of DL-HEPP, an analogue of DL-HEPB, where good anticonvulsant effects were found at the 50 mg/kg dose [26] and anticonvulsant effects like those of valproate at the dose of 100 mg/kg [21]. For its part, the dose of 200 mg/kg was selected as twice the highest dose tested, with the intention of ensuring the observation of the possible toxic effects on development that DL-HEPB could generate. So, the information was added in the corresponding section (lines 139-148).

Comment 5: In the statistical analysis (lines 139-146), was a test to assess the normality of the data performed so that parametric or non-parametric analysis could be performed? This information needs to be described.

Response 5: Thank you for pointing this out. It is correct, the data were analyzed using the Shapiro-Wilk test to determine if they followed a normal distribution. The information has been added in the corresponding section (lines 174-176).

Comment 6: Minor points. On the abstract (line 31), there is an inconsistency "the dose of 400 mg/kg" Shouldn't it be "the dose of 200 mg/kg"?

Response 6: Thank you for your observation; you are right, the correct dose is 200 mg/kg. It was corrected (line 31).

Reviewer 2 Report

The study entitled as “Developmental toxicity study of DL-4-hydroxy-4-phenylhexanamide (DL-HEPB) in rats” is interesting. However, there are several questions that needs to be addressed before final acceptance.

1.    Author must discuss the pathological relevance of the dark red color of liver and spots that they observed in kidney upon treatment with DL-HEPB.

2.    There are several typo errors that was found, which needs to be corrected before submission.

3.    I could not find the description, how authors have quantified the food consumption? And of course, the number of animals kept in a group? How much amount of food was supplemented?

4.    Author must briefly point out the significance of the study.

5.    What is the effective dosage of DL-HEPB as an anti-epileptic drug?

There are several typo errors was found, which needs to be corrected before submission.

Author Response

Comment 1: Author must discuss the pathological relevance of the dark red color of liver and spots that they observed in kidney upon treatment with DL-HEPB.

Response 1. Thanks for your suggestion. We found it very interesting and accurate to explore what could be responsible for the changes in organs observed. The information was added in the corresponding section (lines 264-285).

Comment 2: There are several typo errors that was found, which needs to be corrected before submission.

Response 2: Thank you for your comment. The manuscript has been sent again for style revision to a person associated with the subject of the manuscript. We hope that this time the syntax is adequate.

Comment 3: I could not find the description, how authors have quantified the food consumption? And of course, the number of animals kept in a group? How much amount of food was supplemented?

Response 3: Thank you for your comment. The number of animals per group (20 animals per group) is indicated on line 120. The information has been added in the corresponding section (lines 134-135 and 150-155).

Comment 4. Author must briefly point out the significance of the study.

Response 4: Your comment seems very appropriate to us, so the requested information has been added in the lines 332-343.

Comment 5. What is the effective dosage of DL-HEPB as an anti-epileptic drug?

Response 5: Thank you for your question. DL-HEPB has shown good antiepileptic effects at a dose of 50 mg/kg. While at a dose of 100 mg/kg it has shown effects like those of valproic acid. The information is displayed on lines 139-148.

Reviewer 3 Report

Epilepsy is one of the most common chronic diseases in women of reproductive age. It increases the risk of pregnancy and childbirth complications for mother and child. The effects of prolonged tonic-clonic seizures, during which hypoxia may occur, and the teratogenic effect of drugs pose a threat to the fetus. In addition, the antiepileptic drugs used may interfere with the development of the fetus, causing birth defects in the child. The risk of their appearance is highest in the early stages of pregnancy, when a woman is unaware of her condition. Therefore, it is necessary to think about the health of the child even before pregnancy, choosing medications that have the least impact on the child's development. Therefore, the presented manuscript deals with a very important and topical problem.

This study is timely and scientifically sound and properly written, following all the guidelines for publications of scientific articles and it is within the scope of the journal. The abstract of the manuscript fully covers all parts of the manuscript. The tables additionally facilitate the full evaluation of the content contained in the manuscript and significantly increase its value. The methods are described in a detailed and clear manner. The discussion contains the most important information necessary to draw conclusions from the conducted research.

However, in my opinion the Authors should introduce some corrections and supplement the information:

In the introduction, the Authors should add more information about the harmful effects of all currently used antiepileptic drugs on the fetus.

I suggest adding information on where the idea to use the test compound came from. How should it be better than the currently used ones?

The Authors should add information about the exact mechanism of action of the study drug, this will explain much about the possible effects and benefits of its use.

The Authors should add information how many females and fetuses of rats were used in experiments.

The description of the results lacks exact data on the ANOVA values.

The presented studies are promising, however, as the Authors emphasized, they require continuation to accurately determine the toxic potential of the compound. The Authors should add limitations of the study.

Author Response

Comment 1: In the introduction, the Authors should add more information about the harmful effects of all currently used antiepileptic drugs on the fetus.

Response 1: We appreciate your comment. Although it seems interesting to us, it would be too long to describe the damage that each of the AED that are currently used (only the most used are over 18) exerts on the embryo or fetus. And since several of these AED share a mechanism of action (or are very similar) the damage to the embryo is also very similar, so again it would be very repetitive to try to carry out this description. Therefore, we consider that it is more practical to explain which AEDs are currently most used, their action mechanism, and describe in a general way the main damages that this group of drugs can generate on the intrauterine development. The information is in the lines 58-67.

Comment 2: I suggest adding information on where the idea to use the test compound came from. How should it be better than the currently used ones?

Response 2: Thank you for your suggestion. Currently, the text already describes the origin of the idea to investigate this compound by expressing the need to search for new alternatives of anticonvulsant molecules with fewer toxic effects due to the risk that current AEDs represent for the developing conceptus (lines 75-78). In addition, it is also explained based on previous antiepileptic studies how DL-HEPB has very good antiepileptic effects and less toxicity than traditional AEDs, referring to its novel structure and the different animal models in which it has been evaluated (lines 79-87). However, we have made some adjustments to the text to accommodate your suggestion: Status epilepticus is a clinical problem in medicine. Near 30% epileptic patients exhibit resistance for the treatment. Phenyl alcohol amides are the only anticonvulsant drugs known that protect against the gamma-aminobutyric acid withdrawal syndrome (GWS), a model of epilepsy which is resistant to clinical antiepileptic drugs such as phenytoin, barbiturates, ethosuccimid, valproic acid, carbamazepine, diazepam (the drug choice in cases of status epilepticus), even pentobarbital at anesthetic doses (lines 75-84).

Comment 3: The Authors should add information about the exact mechanism of action of the study drug, this will explain much about the possible effects and benefits of its use.

Response 3: Thank you for your suggestion. We agreed that adding information on the exact anticonvulsant mechanism of DL-HEPB would add much to the manuscript in terms of its potential effects and uses. However, although we would like to do so, it is impossible for us since DL-HEPB is a new molecule and there are no studies that have investigated its mechanisms of action. As we explained in our manuscript, the only data we have on this structure are those referring to its acute toxicity and its pharmacological effects, which were studied in various animal models. However, these studies are only limited to the quantification of its effects. We have not yet carried out studies with which we can demonstrate that the molecule binds to a certain receptor, or to a specific sequence of amino acids that activates or inhibits a certain process... This is precisely the next step in our line of research; elucidate as far as possible the mechanism of action of DL-HEPB.

Comment 4: The Authors should add information how many females and fetuses of rats were used in experiments.

Response 4: Thank you for your recommendation. We have added the information corresponding to the number of male and female rats that were used in each study (lines 108, 121, and 134-135). Regarding the number of fetuses analyzed in the teratogenic study, they are found in tables 2 (No. of live fetuses), table 3 (Total examined fetuses), table 4 (Total examined fetuses) and table 5 (Total examined fetuses).

Comment 5: The description of the results lacks exact data on the ANOVA values.

Response 5: Thank you for your comment. As is commonly done, when designing our experiment we decided to work with an alpha value equal to 0.05, as the significance level. In this context, the importance of the ANOVA lies in finding, or not, values equal or less than our criterion of significance to accept or reject our alternative hypothesis. Thus, we accept our alternative hypothesis both with values of p ≤ 0.05 and with values of p= 0.049. You are right that we do not put exact values when we find statistical differences. However, in the context of the test it is accepted to indicate only if the values are equal or less than 0.05 to express statistically significant differences.

Comment 6: The presented studies are promising, however, as the Authors emphasized, they require continuation to accurately determine the toxic potential of the compound. The Authors should add limitations of the study.

Response 6: Thank you for your correct suggestion. We agree that it is necessary to mention the limitations of this study to better define its scope. The information was added in the corresponding section (lines 332-343).

Round 2

Reviewer 2 Report

The Author's response is satisfactory to me and the revised version has been improved compared to the original submission. Therefore, I decided to accept the manuscript.